# Adopting a WHO Framework Convention on Tobacco Control-Based Tobacco Control Law in Ethiopia: Sustained Transnational Health Advocacy and Multi-Sectoral Institutionalized Support

**DOI:** 10.3390/ijerph21030280

**Published:** 2024-02-28

**Authors:** Eric Crosbie, Kyle Edison, Vandyke Maclean, Dereje Moges, Caroline Fuss, Monique E. Muggli, Bintou Camara Bityeki, Ernesto M. Sebrié

**Affiliations:** 1School of Public Health, University of Nevada, Reno, NV 89557, USA; kedison@nevada.unr.edu (K.E.); kpakpomaclean30@gmail.com (V.M.); 2Ozmen Institute for Global Studies, University of Nevada, Reno, NV 89557, USA; 3Campaign for Tobacco-Free Kids, Washington, DC 20005, USA; dmoges@tobaccofreekids.org (D.M.); cfuss@tobaccofreekids.org (C.F.); mmuggli@tobaccofreekids.org (M.E.M.); bcamara@tobaccofreekids.org (B.C.B.); esebrie@tobaccofreekids.org (E.M.S.)

**Keywords:** tobacco control, tobacco industry, health advocacy, Ethiopia, Africa

## Abstract

Objective: The objective of this study was to document how Ethiopia adopted a WHO Framework Convention on Tobacco Control (FCTC)-based tobacco control law. Methods: We analyzed publicly available documents, including news media articles, advocacy reports, and government documents. We triangulated these findings by interviewing nine key stakeholders. Data were analyzed to construct a historical and thematic narrative and analyzed through a retrospective policy analysis. Results: Local and international health advocacy efforts helped introduce and support WHO FCTC-based legislation by (1) educating policymakers about the WHO FCTC, (2) providing legal assistance in drafting legislation, (3) generating local data to counter industry claims, and (4) producing media advocacy to expose industry activity. Health advocates worked closely with government officials to create a multi-sectoral tobacco committee to institutionalize efforts and insulate tobacco companies from the policymaking process. Japan Tobacco International bought majority shares of the government-owned tobacco company and attempted to participate in the process, using standard industry tactics to undermine legislative efforts. However, with health advocacy assistance, government officials were able to reject these attempts and adopt a WHO FCTC-based law in 2019 that included 100% smoke-free indoor places, a comprehensive ban on tobacco advertising, and large pictorial health warning labels, among other provisions. Conclusion: Sustained local health advocacy efforts supported by international technical and financial assistance can help establish WHO FCTC-based tobacco control laws. Applying a standardized multi-sectoral approach can establish coordinating mechanisms to further institutionalize the WHO FCTC as a legal tool to build support with other government sectors and insulate the tobacco industry from the policymaking process.

## 1. Introduction

Tobacco use remains the most preventable cause of death and disability worldwide, killing more than 8 million people annually, and 80% of these deaths occur in low- and middle-income countries (LMICs) [1]. To address the tobacco epidemic, World Health Organization (WHO) Member States adopted the WHO Framework Convention on Tobacco Control (FCTC) in 2003, and the treaty officially entered into force in 2005. The WHO FCTC places obligations on parties to reduce the supply and demand of tobacco [2], which has helped accelerate the adoption of evidence-based best practices aimed at reducing tobacco use, nicotine addiction (Article 14), and exposure to secondhand tobacco smoke, including smoke-free public places and workplaces (Article 8); pictorial health warning labels (HWLs) (Article 11); and tobacco advertising, promotion, and sponsorship (TAPS) bans (Article 13), among other policies [3,4,5].

While tobacco use began declining in high-income countries in the 1990s and 2000s and in some LMICs beginning in the 2010s [1], the WHO African Region (AFRO) has recently seen a rise in tobacco marketing exposure and usage. Currently, AFRO has the lowest smoking prevalence of all WHO regions (roughly 6% of the world’s smokers), but transnational tobacco companies are aggressively investing and marketing their products, especially to youth and women, in several growing economies in the region [6,7]. Currently, over 77 million adults smoke in AFRO, and this figure is expected to grow gradually over the next century, reaching 413 million smokers by 2100 [8]. Furthermore, the prevalence of smokers in AFRO is anticipated to rise from 15.8% to 21.9% by 2030 based on 2020 levels, the largest projected increase in the world [9].

Given the rapid increase in tobacco marketing exposure and tobacco use prevalence, it is particularly imperative for governments in the WHO AFRO region to develop and implement WHO FCTC-based policies to prevent and reduce tobacco use. These measures include general obligations of parties to protect their public health policies against tobacco industry interference (Article 5.3) and measures aimed at reducing the demand for tobacco products: Articles 6 (price and tax measures), 7 (non-price measures), 8 (protection from exposure to tobacco smoke), 9 (regulation of tobacco product contents), 10 (regulation of tobacco product disclosures), 11 (packaging and labeling), 12 (education, communication, training, and public awareness), 13 (tobacco advertising, promotion, and sponsorship), and 14 (tobacco dependence and cessation). It also includes measures aimed at reducing the supply of tobacco products; Article 15 (illicit trade in tobacco products), 16 (sales to and by minors), and 17 (support for economically viable alternative activities). To date, the majority of the literature studies evaluating the adoption of WHO FCTC-based policies has been conducted in middle- and high-income countries [10,11,12], yet knowledge gaps remain in low-income countries [13,14], particularly those in the WHO AFRO region [7,15,16]. However, the recently published literature has begun to assess policy adoption and implementation of WHO FCTC-based best practices in WHO AFRO, including TAPS restrictions and bans across the region [17], tobacco standardized plain packaging in Mauritius [18], 100% smoke-free environments in Ethiopia [19], large pictorial HWLs in Nigeria and Senegal [20,21], and a shisha ban in Kenya [22].

Despite increased efforts to document and assess WHO FCTC-based policies in WHO AFRO, there still lacks a clear understanding of facilitators and barriers influencing policy adoption. Nowhere is this more evident than in Ethiopia, which is the second most populous country in AFRO, home to over 123 million people. Although Ethiopia is classified by the World Bank as a low-income country and has historically had comparatively low rates of tobacco smoking in the region (2016 tobacco-use prevalence rates at 5.0%; 8.1% men; 1.8% women) [23], the economy is one of the fastest growing in the world and cigarette sales have steadily increased, especially among youth and women [24]. While the previous literature examining tobacco control in Ethiopia analyzed smoking prevalence rates in the country [25,26,27,28], tobacco control prevention efforts [29,30,31], and tobacco industry interference [32,33], this represents the first known study that attempts to understand the process of adoption of a WHO FCTC-based national tobacco control law in Ethiopia. In doing so, this study hopes to provide lessons for other LMICs, especially in WHO AFRO, working to adopt policies aligned with the WHO FCTC and its implementation guidelines.

## 2. Methods

### 2.1. Data Collection

#### 2.1.1. Tobacco Control Legislation

Between March and August 2023, we reviewed Ethiopian tobacco control laws available at Tobacco Control Laws (https://www.tobaccocontrollaws.org/legislation/ethiopia/laws, accessed on 20 May 2023), a database developed by a team of lawyers from the International Legal Consortium at the Campaign for Tobacco-Free Kids (CTFK). The database provides a detailed legal analysis of WHO FCTC-based national tobacco control laws adopted (but not necessarily implemented) globally. Lawyers use standardized review templates and interpretation protocols to analyze tobacco control laws and determine the extent to which the legislation aligns with several articles included in the WHO FCTC and its implementation guidelines.

#### 2.1.2. Reports and News Articles

We also analyzed government and health group reports and news articles by using the search engine Google (https://www.google.com, accessed on 15 April 2023) and Tobacco Watcher (https://tobaccowatcher.globaltobaccocontrol.org/, accessed on 5 June 2023), a surveillance platform that has identified and compiled tobacco-related news from 147 countries in 20 languages, using standard snowball searches [34]. Initial search terms included “Ethiopia”, “tobacco”, “smoking”, and “cigarette”, as well as key dates and specific actors. In total, 23 search terms were used to retrieve 206 documents. We reviewed all of these documents, and 26 documents were deemed relevant for the current study. 

#### 2.1.3. Interviews with Key Informants

Between July and August 2023, we invited 23 key stakeholders via email that had been involved in tobacco control in Ethiopia between 2014 and 2023 to participate in an in-depth interview. Interviewees were identified through media searches, the authors’ networks, and snowball sampling. Nine agreed to be interviewed, four denied our requests, and ten never responded after five requests. The interviewees included three tobacco control advocates, two academic researchers, and four government officials. The interviewees were emailed semi-structured interview questions and agreed to verbal consent to participate in the study in accordance with a protocol approved by the University of Nevada, Reno Committee on Human Research. Interview questions focused on the history of tobacco control in Ethiopia, tobacco industry interference, health advocacy support, and the process of developing and passing Law 1112. Interviews occurred via Zoom and lasted approximately 60–70 min.

### 2.2. Data Analysis

Given that the CTFK Tobacco Control Laws database contains laws already analyzed, we used information available to document the legislative history of tobacco control laws in Ethiopia. This consisted of documenting the key provisions (e.g., smoke-free public places) of each law that align with the WHO FCTC. In analyzing government and health group reports and news articles, we coded these documents based on key events and analyzed them in detail to construct a historical and thematic narrative. Based on these initial findings, we developed semi-structured questions to further understand how the policy was adopted. Results were triangulated and thematically analyzed through standard process tracing frameworks [35] and analyzed through a retrospective policy analysis [31] to generate findings about how the policy was made, identify factors that influenced policymaking, and evaluate past efforts that helped achieve policy goals.

## 3. Results

Law 1112 exemplifies a comprehensive alignment with the WHO FCTC. Key provisions include robust protection of public health policies from tobacco industry interference, notably through a complete ban on the industry’s corporate social responsibility (CSR) activities (Article 5.3). The law establishes a 100% smoke-free environment in all indoor workplaces, public spaces, and public transportation (Article 8); regulates tobacco product contents and disclosures (Articles 9 and 10); mandates 70% pictorial HWLs (Article 11); and introduces comprehensive restrictions on TAPS, including a ban on point-of-sale displays, (Article 13). The law addresses the sale to and by minors by raising the legal age for tobacco sales to and by minors to 21 (Article 16). Furthermore, Law 1112 specifically prohibits the sale of shisha, Electronic Nicotine Delivery Systems, heated tobacco products, and flavored tobacco products, demonstrating its extensive scope in reducing tobacco accessibility and appeal.

The regulation of tobacco in Ethiopia essentially began in the period of 2007–2011, when a series of proclamations mandated that the Ethiopian Food, Medicine, and Healthcare Administration and Control Authority (EFMHACA) become the federal regulatory body responsible for regulating tobacco products (Table 1). This new focus on tobacco control continued in 2014 and 2015, as the Ethiopian government ratified the WHO FCTC on 23 June 2014 and adopted measures attempting to regulate tobacco. Measures included adoption of Ministers Regulation No. 299/2013 in 2014 that required 30% textual and/or pictorial HWLs on tobacco products [36] and Tobacco Control Directive No. 28/2015 in 2015 that established some smoke-free areas in public places and restricted TAPS [37] (Table 1). These efforts set the stage for drafting comprehensive WHO FCTC-based tobacco control legislation.

### 3.1. Assessment of Tobacco Control Progress in Ethiopia (2014–2015)

Following Ethiopia’s ratification of the WHO FCTC, there was a renewed focus on assessing tobacco control in the country. Initial efforts began with a multi-sectoral approach from the government to evaluate gaps in implementing proven WHO FCTC-based measures to reduce tobacco use in Ethiopia. 

### 3.2. Creation of the National Tobacco Control Coordinating Committee

In May 2014, the EFMHACA established a multi-sectoral National Tobacco Control Coordinating Committee (NTCCC) at the national level [30]. The committee included stakeholders from a variety of organizations, including the Ministry of Health, Ethiopian Public Health Institute, Ethiopian Public Health Association (EPHA), Ethiopian Revenue and Customs Authority, Ministry of Labor and Social Affairs, Ministry of Youth and Sports, tobacco control advocacy groups, and professional societies, in order to institutionalize and mainstream tobacco control efforts in various sectors and institutions [30]. This was accomplished with support from WHO Mission experts [38], who helped remind the Ethiopian government that WHO FCTC Article 5.2(a) obliges Parties to strengthen governance for tobacco control and that, “Towards this end, each Party shall, in accordance with its capabilities: establish or reinforce and finance a national coordinating mechanism or focal points for tobacco control” [2]. Key stakeholders interviewed for this study mentioned that this was an important development that helped “institutionalize” tobacco control efforts, which “helped communicate the importance of the WHO FCTC to other government ministries”, such as agriculture and trade [39,40,41,42,43,44,45].

### 3.3. Ethiopian Public Health Institute (EPHA) Trainings

In order to implement the WHO FCTC, the EPHA began educating other members of the NTCCC on the importance of the treaty and its use as a legal tool and basis for assessing tobacco control in the country. Throughout 2015, the EPHA held trainings and workshops with policymakers, government officials, youth, high schools, media, hotels, and other stakeholders [46]. In particular, the EPHA provided training for 25 journalists on the health effects of tobacco smoking and evidence-based measures proven to reduce tobacco use, prepared material for journalists to be used as reference when reporting on tobacco, formed alliances with private- and state-owned media, and provided inputs on how to influence tobacco control [39,41,47]. They also organized two panel discussions which were aired through Ethiopia radio and television and lobbied government officials to adopt WHO FCTC policies [46]. Members of the EPHA reported success in creating awareness among high-level stakeholders by mobilizing the public through media events (e.g., World No Tobacco Day), sharing evidence and data related to tobacco use in Ethiopia, and using training material (e.g., stickers and messages) to educate various stakeholders [46].

### 3.4. Recognition of Tobacco Control Struggles

Despite some advances with smoke-free places and TAPS restrictions, local civil society health advocacy groups such as the Mathewos Wondu-YeEthiopian Cancer Society considered the tobacco control landscape to be limited [39,41,47]. Various health advocates mentioned during interviews for this study, as well as to the media in 2015, that the National Tobacco Enterprise (NTE), the government-owned tobacco company, had continued to violate smoking and advertising restrictions [39,41,47]. For example, NTE continued placing large billboards at the city center in the capital city, Addis Ababa, and placing advertisement posters on busy streets, making them easily visible to pedestrians and passing traffic; these were prominent particularly in poor areas of the capital [48]. Various health advocates reported that children continued selling single cigarette sticks on the streets [48], with one advocate stating, “children as young as ten years old can be spotted daily selling cigarettes in downtown locations” [39]. Health advocates admitted that, at the time, there was a common understanding that their efforts were limited in many areas, including (1) resources; (2) the coordination and cooperation with national and international organizations; (3) international experiences and support; (4) the capacity of enforcement mechanisms, structure, and infrastructures; and (5) evidence (both synthesized data and the generation of data) [46]. As a result, health advocacy groups recognized the need for more focused and targeted advocacy strategies and campaigns aimed at tobacco companies and the need for the government to adopt WHO FCTC-based legislation [46].

### 3.5. Health Advocacy Efforts to Introduce WHO FCTC-Based Legislation (2015–2016)

While the EPHA was making important advancements regarding educating stakeholders about the importance of tobacco control, local health advocacy groups, with the support from international and regional partners, began building upon these efforts by using targeted advocacy strategies [39,41,47]. Following the EPHA’s journalist training sessions, health advocates noticed that journalists had established a coordinating committee network, consisting of members from different media institutions [46]. Advocates reported noticing that media coverage on tobacco control in Ethiopia had increased in both national electronic and print media (local newspapers and television and radio programs) [46]. Despite the EPHA’s meetings, workshops, and training of 25 journalists representing different media, health advocates felt that journalists’ understanding of the health, economic, and social impact of tobacco remained low [46]. Furthermore, there was the belief that media participation in tobacco control campaigns was generally low and responsive to issues only when a workshop was held or a press statement was issued [46]. A reflection from advocates was that the number of trainings held compared to the number of journalists actively involved in or contributing to tobacco control media was not adequate, and that media, although improved, was not well organized enough to push WHO FCTC-based legislation [46]. As a result, health advocacy groups focused their efforts on further engaging with journalists and creating a deeper and well-coordinated network of journalists who would be trained to constantly engage the government and the public on tobacco control issues [46]. This included training 50 journalists on the magnitude, perceived risk factors, and the effects of tobacco smoking and measures to prevent tobacco initiation and reduce use [46]. Several of the interviewees suggested that the media was more favorable to tobacco control following the training workshops [39,41,42,43,49].

Between June and August 2016, health advocacy groups held a series of workshops to further engage and educate various stakeholders on tobacco control-related issues. On 20 June 2016, health advocates held a workshop and invited federal government officials from various ministries, including the Ministries of Finance, Education, and Transport, among others, in an open forum to discuss and exchange ideas on existing legal frameworks on tobacco control, gaps in current tobacco control policies in Ethiopia, and related provisions of the WHO FCTC and its implementation guidelines [50]. Advocates, along with lawyers and academics, presented evidence on the harmful effects of tobacco advertising, especially to youth; how designated smoking areas do not protect against secondhand smoke exposure; tobacco industry interference; and policy frameworks to help introduce WHO FCTC-based legislation [50]. On 24 June 2016, health advocates held a workshop and invited state policymakers and city administration-level regulators to discuss and exchange ideas on the best regional experiences, legal possibilities for regional states to issue stronger tobacco control laws, and key related provisions of the WHO FCTC and its implementation guidelines [50]. On 27 June 2016, health advocates held another workshop and invited various media members to help familiarize them with the main issues in tobacco control laws, discuss how the media could be helpful to strengthen the implementation of tobacco control laws, and help bring about policy changes on key issues [50]. On 1 and 8 August 2016, health advocates held two final workshops and invited policymakers, lawyers, and health regulators to train them on the WHO FCTC, its implementation guidelines, and federal and state laws on tobacco control [50].

In addition to educational workshops and trainings, the EFHMACA, the EPHA, and the WHO country office administered the Global Adult Tobacco Survey (GATS), a nationally representative household survey designed to collect nationally representative data on Ethiopians aged 15 years and older [23]. While the 2016 GATS data indicated low tobacco use prevalence rates, 29.3% (6.5 million) of adults (aged 15+) in Ethiopia reported exposure to secondhand smoke when at the workplace, more than 20% of adults reported noticing anti-cigarette smoking information on the TV or radio, and 23.3% of current smokers thought about quitting because of a warning label [23]. All of the interviewees mentioned that this was significant local data that illustrated the need to further advance tobacco control and adopt WHO FCTC-based legislation [39,40,41,42,43,44,45,47,49].

### 3.6. Introduction of Draft Proclamation 1112 (2015–2017)

Throughout the mid-2010s, tobacco control advocates continued to work with government officials to develop tobacco control legislation. Despite political instability, advocates were able to help institutionalize a multi-stakeholder approach to finally introduce WHO FCTC-based legislation.

### 3.7. Political Instability

Due to political grievances and civil unrest, on 9 October 2016, the Ethiopian Prime Minister at the time, Hailemariam Desalegn, declared a state of emergency, imposing restrictions on freedom of speech and access to information. This caused massive protests by ethnic groups against the government and disrupted legislative activity due to frequent cabinet reshuffling. All of the key stakeholders interviewed for this study admitted that this disrupted and delayed the progression of WHO FCTC-based legislation, as other matters were given priority [39,40,41,42,43,44,45,47,49]. However, health advocates had laid the foundation of close relationships with government officials and continued developing potential draft language for WHO FCTC-based legislation despite these delays [39,40,41,42,43,44,45,47,49]. As one health advocate recounted, “we established such strong relationships with the EFDA that they trusted us to help develop draft legal language” [41]. Equally as important, health advocates were able to educate EFHMACA officials on the importance of WHO FCTC Article 5.3, which requires WHO FCTC parties to protect their public health measures from the commercial interests of the tobacco industry. All of the interviewees claimed that this was vital to creating strong evidence-based draft language and preventing any industry interference during the drafting and development of WHO FCTC-based legislation [39,40,41,42,43,44,45,47,49].

### 3.8. National Tobacco Control Strategic Plan (2017–2020)

The creation of the NTCCC in 2015, increased pressure from health advocates, and proactive government activity served as a foundation for the creation of the National Tobacco Control Strategic Plan (2017–2020), which culminated in October 2017 [30]. The Strategic Plan aimed to guide program managers, public health professionals, partners, and stakeholders in their efforts to plan and implement tobacco control strategies that aligned with the WHO FCTC [30]. The Strategic Plan also helped institutionalize this process by again taking a multi-sectoral approach involving various government ministries to reduce tobacco prevalence by 15% by the end of 2020 and achieve a “tobacco-free Ethiopia”. The Strategic Plan recognized potential weaknesses in the plan to obtain these goals (e.g., inadequate data and evidence on the social and economic impact of tobacco use) and identified potential threats from the tobacco industry (e.g., growing industry interference) [30]. As a result, the Strategic Plan recognized the importance of building capacity in terms of resources, expertise, and stakeholder collaborations and established nine strategic objectives and 24 strategies to implement the WHO FCTC [30].

Following continuous discussions and drafting of WHO FCTC-based legislation with health advocacy groups and various government ministries, the EFMHACA introduced a draft of the Food and Medicine Administration Proclamation 1112 in July 2017. The draft proclamation included various provisions related to public health, including provisions addressing alcohol control and medicine; its tobacco control provisions centered on establishing 100% smoke-free indoor workplaces and public places, banning TAPS, and requiring pictorial HWLs, among others. When asked about the inclusion of tobacco control measures in an omnibus type of legislation rather than a standalone tobacco control draft bill, interviewees from academia, civil society, and government mostly agreed that this was a strategic choice based on the lack of resources and political environment of the country [39,40,43,45,47,49]. A few interviewees claimed, “this was a realistic move” and that “without including various other health measures, the executive cabinet would have never moved forward with the legislation” [40,43,44,45].

### 3.9. Increased Tobacco Industry Presence and Involvement (2016–2018)

On 19 May 2016, Japan Tobacco International (JTI) offered USD 510 million, the highest government sale of a public enterprise in Ethiopia, to acquire 40% of the government-owned tobacco company, NTE Share Company [51]. On 21 December 2017, JTI became the majority shareholder of NTE by spending USD 424 million to increase its share from 40% to 71% [52]. Following these acquisitions, JTI’s CEO asserted, “Ethiopia will be an important expansion of our geographic footprint in emerging markets”, citing “double-digit economic growth” and expectations that volumes would continue to rise [53]. The CEO went on later to state, “The significant increase in our ownership of NTE shares reaffirms our strong belief in the company and Ethiopia as an increasingly important place to do business in Africa” [52].

### 3.10. Memoranda of Understanding

As JTI acquired shares of NTE, it also began negotiations in 2017 with the Ministry of Public Enterprises and the Ethiopian Revenue and Customs Authority for further involvement in tobacco affairs. On 17 December 2017, NTE sent a letter to the Ministry of Public Enterprises and copied various other ministries in government, reminding them that the Ethiopian government had committed to “continued investment in the manufacturing, agricultural, distribution and sales operations of NTE” [54]. In return, NTE said that it remained committed to supporting the government’s fight against illicit tobacco trade and that the government “should use its best efforts to consider and support NTE’s recommendations and investment in this regard” [54]. As a result, NTE argued that all relevant industries were expected to complete a Memorandum of Understanding (MoU) with NTE to reduce illicit tobacco trade [55]. In the letter, NTE claimed that “the Government’s anti-illegal trade task force and NTE should collaborate closely to define an action plan and a list of measures to best ensure such objective can be achieved” [54]. NTE went on further to state that it should be consulted in a transparent manner during the law-making process and that the government should “take all measures necessary to ensure that NTE retains its monopoly rights until 2025, as per the Contract of sale signed on 15 July 2016” [54].

### 3.11. Public Health Advocacy Warns about JTI’s Acquisition of NTE 

On 4 January 2018, tobacco control advocates sent a letter to the Health Minister Yifru Birhan to discuss JTI’s recent investment in acquiring NTE [56]. In the letter, health advocates detailed their concern about JTI’s acquisition and that this would “undoubtedly bring a challenge” to the Health Ministry and EFMHACA [56]. However, health advocates continued to express their “continued full support, technically and financially, in defending the integrity of Ethiopia’s public health policymaking from the industry interference” [56]. The letter went on to state that the draft proclamation was legally defendable and that the industry’s attempts to interfere with the policymaking process were not unique to Ethiopia [56]. The letter also stressed that health advocates had supported governments around the world for over a decade in opposing the industry’s attempt to “hijack policymaking, influence political opinion, offer to participate in the drafting of laws which is contrary to WHO FCTC, and compromise implementation of the WHO FCTC”, and that, in almost all cases, health advocates had come out victorious, including in Kenya, Uganda, and Nigeria [56]. The letter concluded by stating that these health advocacy groups wished to pledge their “continued technical and financial support to the efforts to help towards this important accomplishment” [56]. This included mentioning that they were in the process of “funding a study to quantify the amount of illicit trade to counter industry arguments, which undoubtedly exaggerated the levels of illicit trade-a scare tactic often used despite JTI itself having been complicit in cigarette smuggling” [56].

Additionally, health advocates provided information to the EFMHACA assessing JTI’s acquisition in relation to international trade and investment agreements. On 20 January 2018, health advocates sent the EFMHACA an assessment letter warning government health officials to be aware of trade arguments, as tobacco companies seek to delay and prevent proposed tobacco control laws by alleging that laws breach international obligations [39,41,47]. The assessment letter outlined that allegations may be resisted by minimizing the risk of a potential industry claim [39,41,47]. Advocates recommended that EFMHACA follow appropriate constitutional and administrative due processes when adopting new tobacco control measures; otherwise, tobacco companies could claim that they were not granted “fair and equitable treatment” under many international investment agreements. Advocates further stated that if a claim were to arise, international technical and legal support was available to assist LMICs, including financial support through the Bloomberg/Gates Anti-Tobacco Trade Litigation Fund [57]. Advocates assessed Ethiopia’s international trade and investment agreements and claimed that even though Ethiopia had bilateral investment treaties with the Netherlands and Switzerland, where JTI, as a transnational corporation, had holdings as a foreign investor, likely investment clauses (e.g., expropriation, fair and equitable treatment) would be rejected based on similar provisions already rejected in investment challenges to Australia and Uruguay’s tobacco packaging and labeling policies [58,59].

### 3.12. Public Consultation Sessions for Draft Proclamation 1112

In August 2017, the EFMHACA held open public consultations for 30 days for all stakeholders to provide comments either in person or electronically in reference to the draft proclamation. Supporters of the proclamation included the Mathiwos Wondu-YeEthiopia Cancer Society; Ethiopian Thoracic Society; Health, Development, and Anti-Malaria Association, Ethiopia; CTFK; African Tobacco Control Alliance (ACTA); and individual health advocates, academics, and lawyers. Opponents included the tobacco industry (e.g., NTE), commercial associations (e.g., Ethiopian Chamber of Commerce), and tobacco farmers. The majority of comments focused on the areas of (1) illicit tobacco trade; (2) tobacco packaging and labeling; (3) tobacco advertising, promotion, and sponsorship (TAPS); (4) smoking in public places; and (5) minimum pack size (Table 2). The following is a summary of stakeholder positions on these provisions and responses from the EFMHACA, which overwhelmingly sided with supporters.

#### 3.12.1. Illicit Tobacco Trade

Opponents led by the NTE argued that illicit trade remained a serious problem, accounting for high amounts of smuggled cigarettes into the country, especially in the east, from Somalia [60]. NTE also suggested that the EFMHACA needed to succeed against illicit trade before enacting the draft proclamation. Supporters anticipated these industry claims by arguing that it was a standard industry tactic to exaggerate the problem of illicit trade [61]. Advocates commissioned an independent study that eventually showed only 19% of the collected empty cigarette packs were classified as illicit compared to the industry’s exaggerated figure of 61% [62]. Supporters also argued that the industry was compliant in smuggling and demonstrated from other countries that policies such as pictorial HWLs do not increase illicit tobacco trade [39,41,47]. The EFMHACA responded to public comments stating that lifesaving tobacco control measures are more important than illicit tobacco trade [63]. They also reiterated many of the health-advocacy supporting arguments and stated that the WHO Protocol to Eliminate Illicit Trade in Tobacco Products is the best way to address illicit trade [63].

#### 3.12.2. Tobacco Packaging and Labeling

Opponents argued that there was no evidence showing that large and picture-based health warnings increased public awareness, that the harms of tobacco use and exposure to tobacco smoke were incorrect, and that the government should maintain 30% text-only health warnings on cigarette packs [60]. Public health supporters offered international evidence showing that effective health warning labels increase knowledge of risks associated with smoking and influence future decisions about smoking [39,41,47]. In particular, they relied on independent and credible studies showing that large pictorial HWLs motivate smokers to quit, discourage people from starting, and keep ex-smokers from relapsing, and that pictorial warnings are more accessible for people that are illiterate and children, two important and vulnerable populations [39,41,47]. The EFMHACA responded by stating that including large pictorial HWLs on tobacco product packaging was an essential measure aimed at reducing and preventing tobacco use [63]. Furthermore, they stated that the effectiveness of pictorial HWLs increases with size and should aim to cover as much of the principal display area. Thus, the 30% text-only warning fails to raise awareness and thereby fails to protect citizens [63].

#### 3.12.3. Tobacco Advertising, Promotion, and Sponsorship (TAPS) 

Opponents argued that these provisions would not change smoking behavior, especially among minors, and that there was no legal basis or justification for banning TAPS [60]. Specifically, opponents argued that the display of tobacco products at the point of sale allows for genuine competition between manufactures, and banning product display facilitates illicit trade by creating an “under-the-counter culture” [60]. Supporters offered evidence showing how tobacco companies had directly targeted and marketed to youth in Ethiopia, primarily at and around schools [39,41,47]. Supporters showed evidence that tobacco companies had already violated several TAPS restrictions from previous legislation and offered evidence showing that TAPS bans work in other countries by reducing the exposure of tobacco content, especially among youth [39,41,47]. EFMHACA stated that restrictions or bans on only some forms of tobacco marketing have a limited effect, exposure to tobacco products increases tobacco purchasing behavior and smoking status, and tobacco products placed near candy and children’s items and at children’s eye level encourage children to see them as harmless everyday items [63]. Furthermore, the EFMHACA rejected the industry’s notion about crime, stating that banning retail cigarette advertising does not affect the ability of enforcement officials to identify black market sellers or legal businesses selling contraband [63].

#### 3.12.4. Smoke-Free Public Places

Opponents argued that it was the responsibility of smokers to know when and where they should smoke [60]. They also argued that tobacco smoke is highly diluted in outdoor environments and that banning smoking 10 m from a public place will make it very difficult for smokers to smoke [60]. Supporters offered evidence from several countries around the world showing that there was no safe level of tobacco smoke exposure and that adopting such policies contributed to a denormalization of tobacco, thereby helping smokers quit [39,41,47]. The EFMHACA responded to these comments by reinforcing the fact that there is no safe level of tobacco smoke exposure and that designated smoking areas do not protect people from exposure to tobacco smoke [63]. They also stated that tobacco smoke exposure levels can be significant outdoors, particularly when smokers are in close proximity to others [63]. Furthermore, they mentioned that smoke-free outdoor spaces help smokers who are trying to quit, eliminate the sight and smell of tobacco smoke, and contribute to changing social norms in communities by reinforcing to children and youth that smoking is not an acceptable behavior [63].

#### 3.12.5. Minimum Pack Size

Opponents argued that requiring a minimum pack size was discriminatory, hindered innovation, and would cause an inconvenience for smokers that purchase single cigarettes more than packs of cigarettes [60]. Supporters provided evidence that children were selling loose cigarettes on the streets, helping enable the sale and spread of cigarettes, a common problem in LMICs. Supporters provided evidence from other countries showing that this was a successful approach in minimizing youth initiation. The EFMHACA responded by arguing that single cigarettes and small pack sizes allow vulnerable populations such as youth to buy cigarettes without paying the full price, and that youth are more likely to experiment with single cigarettes [63]. The EFMHACA also stated that lower price points undermined efforts to decrease the affordability of tobacco products and that smokers who would otherwise quit may continue to smoke with this access, especially youth [63].

### 3.13. Passage of the Food and Medicine Administration Proclamation 1112 (2019)

On 28 February 2019, the Ethiopian government gazetted the Food and Medicine Administration Proclamation 1112. Law 1112 maintains a majority of the same language included in the original drafts of the bill. It also implements key WHO FCTC-based measures aimed at reducing the demand for tobacco products, including the adoption of 100% smoke-free environments in indoor public places, workplaces, and public transport; a comprehensive ban of all forms of TAPS; a requirement of pictorial HWLs covering 70% of the principal display area of all tobacco packages; tobacco product regulation; and a ban on the sale of waterpipe tobacco, HTPs, and e-cigarettes. 

## 4. Discussion

The process of adopting Law 1112 in Ethiopia contains several important lessons for other countries, especially LMICs and particularly those in AFRO seeking to adopt WHO FCTC-based tobacco control policies. First, WHO FCTC legal obligations helped accelerate policy development and continued to be vital throughout the legislative process in securing Law 1112. Similar to other experiences in LMICs [11,12,13,64], health advocates were able to leverage the WHO FCTC as a legal instrument to pressure government officials into adopting WHO FCTC-based legislation. 

Another important factor of the WHO FCTC helping to drive the development and eventual adoption of Law 1112 was the health advocacy involved in including an institutionalized multi-sectoral approach. Health advocates were able to access and establish close relations with national government officials in the Health Ministry and EFDA, which helped create a multi-sectoral national committee in order to institutionalize and mainstream tobacco control efforts in various sectors and institutions and establish a national coordinating mechanism for tobacco control. Similar to other successful approaches in LMICs, health advocates, along with the Health Ministry and EFDA, were able to communicate the importance of the WHO FCTC as a legally binding treaty to other government ministries, most notably the Agriculture and Trade Ministries. This included stressing the importance and controlling not only active but passive smoking [65,66]. Thus, in some respects, tobacco control advocates were able to engage deeply with the Health Ministry and EFDA, allowing for “bottom-up” innovation to build support with other sectors through a Whole-of Government (WoG) approach [33].

Despite this inter-ministry cooperation, there continues today to be a lack of awareness of WHO FCTC Article 5.3 by other government ministries both in terms of the law’s provisions and engagement with industry executives as stakeholders. While this did not seem to impact the passage of Law 1112, it appears to have weakened and delayed attempts at increasing tobacco taxes (Article 6) in Ethiopia. Consistent with other studies in Ethiopia [32,33], as well as in India [67], there appears to be contrasting levels of awareness of WHO FCTC Article 5.3 across ministries, as several interviewees believed that government officials in the Health Ministry and EFDA were aware of Article 5.3, whereas others were not. This lack of knowledge resulted in the Ethiopian Revenue and Customs Authority signing an MoU with NTE to address illicit tobacco trade. Given that tobacco companies employ voluntary MoUs as a global strategy in several LMICs to undermine efforts to tackle illicit tobacco trade [68], it is imperative that countries like Ethiopia adopt and implement the Protocol to Eliminate Illicit Trade in Tobacco Products, an international treaty adopted in 2012 and ratified by 68 countries as of December 2023, that aims to eliminate all forms of illicit tobacco trade by adopting cooperative cross-border measures [69].

The Ethiopian case study is another example of the importance of coordinated transnational health advocacy efforts. Like other experiences in LMICs [13,16,59,70], the transnational tobacco control network, comprising local health academics and advocates and regional and international health organizations, was able to pool resources to actively confront industry opposition and work closely with policymakers to develop and eventually adopt a WHO FCTC-based law in Ethiopia. In particular, public health advocates were able to coordinate and execute activities at opportune moments. Educating policymakers about the WHO FCTC and providing legal assistance during the early stages of the drafting process resulted in keeping the industry insulated from the process and helped develop close ties with government officials who were then positioned later to reject industry claims. The training of journalists helped create a deeper and well-coordinated network of local journalists that were able to constantly engage the government and public on tobacco control issues. As pressure mounted from the industry concerning complaints about the alleged rise in smuggled cigarettes, advocates were able to produce local data measuring illicit tobacco trade in Ethiopia to demonstrate that the industry and its allies exaggerated the problem. As a result, this policy framing played an important role in passing Law 1112 and securing support, unlike in South Africa, where a lack of support led to opposition against a tobacco ban during COVID-19 [71].

The Ethiopian case study is an example of how governments can adopt WHO FCTC-based policies during political instability. Similar to countries like Nepal [71], Pakistan, Nigeria [72], and Gambia, health advocates in Ethiopia had to navigate political turmoil, civil unrest, and industry interference that caused unexpected delays and posed unique challenges to approving a tobacco control law. In particular, the Ethiopian experience was similar to that of other countries experiencing political volatility in which reforms were successful when an issue was kept “alive” in the community by “policy entrepreneurs” continually repackaging an issue [73]. Health advocates in Ethiopia were able to act as “policy entrepreneurs” amidst instability by training journalists to keep the issue of tobacco control alive and at the forefront in the media, navigating legal complexities of state ownership in the tobacco industry and industry threats of trade obligations and continuing to coordinate with diverse stakeholders through a WoG approach. Conducting a nationwide overview of smokers’ smoking habits (e.g., topography) could also lead to more reasonable and applicable legalization, which may not be reflected immediately but in the long term. Thus, sustained efforts could be replicated in similar contexts to offset instability and adapting in a changing political landscape to adopt best practices.

## 5. Limitations 

Although we reached out to 23 individuals, only 9 agreed to be interviewed. However, a strength of this paper is the diversity of viewpoints, as interviewees included representatives from academia, advocacy, and government. These interviewees were also integrally involved in the policy process and provided an added level of insight to help contextualize our document findings. 

## 6. Conclusions

Sustained local health advocacy efforts supported by international technical and financial assistance can help establish WHO FCTC-based tobacco control laws. Applying a standardized multi-sectoral approach can establish coordinating mechanisms to further institutionalize the WHO FCTC as a legal tool to build support with other government sectors and insulate the tobacco industry from the policymaking process.

## Figures and Tables

**Table 1 ijerph-21-00280-t001:** History of tobacco control legislation in Ethiopia (1999–2023).

Effective Date Going into Force	Legal Measure	Content of the Provisions
23 July 2007	Broadcasting Service Proclamation No. 533/2007	Empowers the Broadcast Authority to license broadcasters (TV and radio), print media, and advertising agencies, among other entities, and enforces compliance with tobacco advertising violations, along with the Ethiopian Food, Medicine, and Healthcare Administration and Control Authority (EFMHACA).
13 January 2010	Food, Medicine and Health Care Administration and Control Proclamation 661/2009	Mandates EFMHACA with the power and responsibility to regulate tobacco products, along with food, medicine, medical devices, traditional medicine, complementary and alternative medicines, cosmetics, health professionals, and health institutions.
27 August 2012	Advertisement Proclamation 759/2012	Mandates the Ethiopian Broadcast Authority (EBA)’s authority over tobacco advertising violation issues, including examining and suspending any advertisement—other than an outdoor advertisement—disseminated in violation of the provisions of the Advertisement Proclamation.
24 January 2014	Food, Medicine, and Health Care Administration and Control Council of Ministers Regulation No. 299/2013	Requires 30% text and/or pictorial warnings on tobacco product packs.
17 February 2014	WHO FCTC ratification Proclamation 822/2014	Requires the Ethiopian government to adopt and recommends that it adopts tobacco control laws based on the WHO FCTC to control the supply and demand of tobacco use.
21 April 2015	Tobacco Control Directive No. 28/2015	Governs, among other things, smoking restrictions, tobacco advertising, promotion and sponsorship, tobacco packaging and labeling, and tobacco product regulation. This directive was later repealed by Tobacco Control Directive No. 771/2021.
28 February 2019	Food and Medicine Administration Proclamation No. 1112/2019	Governs, among other things, smoke-free environments; tobacco advertising, promotion, and sponsorship; tobacco packaging and labeling; tobacco product regulation; protection against tobacco industry interference; and tobacco-related licensing and sales. It also bans the use and sale of e-cigarettes and heated tobacco products.
10 September 2019	Tobacco Products Pictorial Health Warning and Labeling Directive No. 44/2019	Contains detailed requirements related to packaging and labeling.
13 April 2021	Tobacco Control Directive No. 771/2021	Provides implementing provisions for (1) the licensing of manufacturers, importers, and distributors; (2) mandatory disclosure and reporting requirements; (3) inspection and testing of constituents; (4) comprehensive ban on tobacco advertising, promotion, and sponsorship; (5) smoke-free provisions; (6) administrative and legal measures against non-complying products and activities; and (7) the establishment of national Tobacco Control Coordinating Body to ensure implementation of government duties by non-health sector authorities.

**Table 2 ijerph-21-00280-t002:** Draft Proclamation 1112, industry and health-advocate public comments, and government responses.

Concentration Area/Provision	Industry Comments	Health Advocate Comments	Government Responses
Illicit tobacco trade (WHO FCTC Article 15)	- Illicit trade remains a serious problem, accounting for high amounts of smuggled cigarettes into the country, especially from the east, from Somalia - Need to succeed against illicit trade before enacting the draft proclamation	- Standard industry tactic to exaggerate the problem of illicit trade - Independent study shows only 19% of the collected empty cigarette packs were classified as illicit compared to the industry’s exaggerated 61%	- Lifesaving tobacco control measures are more important than illicit trade - Tobacco packaging and labeling restrictions do not impact illicit trade - Tobacco companies, including JTI, have been complicit in cigarette smuggling for decades - More important to strengthen law enforcement and customs and tax administration rather than weakening health proposals - WHO Protocol to Eliminate Illicit Trade in Tobacco Products is the best way to address illicit tobacco trade - Not aware of independent assessment on scale of illicit tobacco - Higher tobacco tax shares in other countries still have much less illicit tobacco trade
Tobacco packaging and labeling (WHO FCTC Article 11)	- There is no evidence showing that larger and picture-based health warnings increase smokers’ awareness - The harms of tobacco use and exposure to tobacco smoke are incorrect - Maintain 30% text-only health warnings on cigarette packs	- Effective health warning labels increase knowledge about risks associated with smoking - Health warnings influence future decisions about smoking - Large pictorial warning labels motivate smokers to quit, discourage from starting, and keep ex-smokers from relapsing - Pictorial warnings are better for low-literacy audiences and children, two vulnerable population groups	- Large pictorial health warnings on tobacco product packaging is an essential aimed at reducing and preventing tobacco use - The effectiveness of health warnings increases with size; should aim to cover as much of the product as possible with health warnings - 30% text-only warning fails to raise awareness and thereby fails to protect citizens
Tobacco advertising, promotion, and sponsorship (WHO FCTC Article 13)	- Display of tobacco products at points of sale facilitate illicit trade by creating an “under-the-counter culture” - TAPS ban will not change smoking behavior - Packaging plays no role in minors’ or adults’ smoking - There is no legal basis or justification - Product display allows for genuine competition between manufacturers	- Tobacco companies have already violated several TAPS restrictions from previous legislation - TAPS bans work in other countries by reducing the exposure of tobacco content, especially among youth	- Restrictions or bans on only some forms of tobacco marketing have limited effect - Absence of a complete ban; tobacco companies will shift their vast resources - Tobacco products are placed near candy and children’s items and at children’s eye level, encouraging children to see them as harmless everyday items - Exposure to tobacco products increases tobacco purchasing behavior and smoking status - No independent and credible evidence that shows an increase in youth smoking or tobacco smuggling is associated with display bans - Absence of retail cigarette displays reduces impulse purchases - Banning retail cigarette advertising does not affect the ability of enforcement officials to identify black market sellers or legal businesses selling contraband
Smoking in public places (WHO FCTC Article 8)	- An extensive smoking ban in areas is not the solution - It is the responsibility of the smokers to know when and where they should smoke - Tobacco smoke is highly diluted in outdoor environments - Banning smoking within 10 m of a public place will make smoker unable to smoke	- Several countries around the world show there is no safe level of tobacco smoke exposure - Adopting such policies contributed to a denormalization of tobacco, thereby helping smokers quit	- There is no safe level of tobacco smoke exposure - Designated smoking areas do not protect people from exposure to tobacco smoke, as smoking easily moves from the smoking areas to areas in a venue where smoking is not allowed - Tobacco smoke exposure levels can be significant outdoors, particularly when smokers are in close proximity to others - Smoke-free outdoor spaces help smokers who are trying to quit, eliminate the sight and smell of tobacco smoke, and contribute to changing social norms in communities by reinforcing to children and youth the idea that smoking is not an acceptable behavior
Minimum pack size (WHO FCTC Article 11)	- Requiring a minimum pack size on tobacco products is discriminatory - It hinders innovation - It discriminates against heated tobacco products - Ethiopian citizens purchase singles more than multiples, so this would cause an inconvenience, and they would seek cheaper alternatives	- Children are selling loose cigarettes on the streets that enable the sale and spread of cigarettes, a common problem in LMICs - Other countries have been successful in using this approach to minimize youth initiation	- Single cigarettes and small packs allow vulnerable populations, such as youth, to buy cigarettes without paying full price - Lower price points undermine efforts to decrease the affordability of tobacco products through tax - Smokers who may otherwise quit because of affordability issues may continue to smoke - Youth access single cigarettes more easily, which may encourage youth non-users to experiment with smoking - Individual cigarettes are displayed, sold, and consumed without consumers being exposed to the large pictorial health warnings

FCTC: Framework Convention on Tobacco Control. LMICs: low- and middle-income countries. TAPS: tobacco advertising, promotion, and sponsorship. WHO: World Health Organization.

## Data Availability

Data is unavailable due to privacy and ethical restrictions.

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
