# Peer review of "Adopting a WHO Framework Convention on Tobacco Control-Based Tobacco Control Law in Ethiopia: Sustained Transnational Health Advocacy and Multi-Sectoral Institutionalized Support"

_ijerph, 2024, doi:10.3390/ijerph21030280_

Round 1

Reviewer 1 Report

Comments and Suggestions for Authors

This work presents the adaptation of WHO FCTC in Ethiopia. Sharing such experiences is globally very useful and important. The paper is informative and flows very well. I have a few suggestions, which I hope to be useful to the authors.

1. The article 3 of the FCTC states the objective. In this article, we read “The objective of this Convention and its protocols is to protect present and future generations from the devastating health, social, environmental and economic consequences of tobacco consumption and exposure to tobacco smoke…” Following this article, it is important to consider both active and passive exposure to tobacco smoke for different types of cigarettes. The authors are welcome to review the published works below (sorted based on publication year) and make a discussion on controlling passive exposure to tobacco smoke by making suitable policies.

2023: https://doi.org/10.1007/s11869-023-01435-9

2022: https://doi.org/10.3390/ijerph19106275

2. Article 6 of the FCTC states the “Price and tax measures to reduce the demand for tobacco.” Local legalization in this case should be done carefully, especially when it comes to e-cigarettes. As an interesting case, I wish to introduce the authors to an incident in Korea. Since 2016, e-cigarette companies have run away from paying taxes by stating that under Korea’s tobacco laws, nicotine extracted from tobacco stems and roots is not classified as tobacco! (Read more here: https://tobaccoreporter.com/2023/09/07/korea-vape-companies-evading-taxes/). Discussing such cases and stating how to avoid them in Ethiopia can engage the readers more in this work.

3. Having a nationwide overview of smokers’ smoking habits (topography) can lead to more reasonable and applicable legalization. This cannot be reflected right away in this work but can be discussed in the discussion about the required (potential?) actions.

4. The authors are encouraged to provide more information about their conducted interview. Presenting the interview questions, along with the sociodemographic information of the interviewees can be helpful to readers from other countries.

In conclusion, I found this work very valuable, and suitable to be considered for publication in IJERPH, after revisions suggested above.

Reviewer 2 Report

Comments and Suggestions for Authors

The manuscript effectively outlines the implementation of National Tobacco Control Laws in Ethiopia. However, there is an opportunity for improvement in clarifying the adoption of the WHO FCTC.

1.      The study identifies specific priority areas outlined in Draft Proclamation 1112; however, a clearer connection with the WHO FCTC is needed. To enhance clarity, it is recommended to explicitly mention the relevant FCTC articles, preferably in the introduction. By establishing this connection early on, readers will gain a comprehensive understanding of how national priorities align with specific WHO FCTC provisions, contributing to a more cohesive and informative manuscript. Incorporating this clarification in the introduction will set the stage for a more seamless exploration of the interconnectedness between national priorities and the corresponding WHO FCTC articles throughout the study. For example In line 37:

·         “The WHO FCTC places obligations on Parties to reduce the supply and demand of tobacco,  which has helped accelerate adoption of evidence-based best practices aimed at reducing tobacco use, nicotine addiction (Article 14) and exposure to secondhand tobacco smoke (Article 8)  including smoke-free public places and workplaces, pictorial health warning labels (HWLs) (Article 11) , and tobacco advertising, promotion and sponsorship (TAPS) bans (Article 13) , among other policies.

·         You can also mention it in this way wherever it is relevant. For example in line 481:

“Despite this inter-ministry cooperation, there continues today to be a lack of aware- 481 ness of WHO FCTC Article 5.3 by other government ministries both in terms of the law’s 482 provisions and engagement with industry executives as stakeholders. While this did not 483 seem to impact the passage of Law 1112, it appears to have weakened and delayed at- 484 tempts at increasing tobacco taxes (Article 6) in Ethiopia.”

·              You can make Table 2 clearer by adding the article number in front of the concentration area ex: Illicit tobacco trade (Article 15)

2.      There is a need for more specificity regarding the incorporation of individual FCTC articles in the study. A summary of how many articles are addressed while incorporating national tobacco control laws in Ethiopia will give clarity on the level of adoption.

3.      The alignment of national laws with the WHO FCTC is not clearly delineated. Further clarification on the implementation of specific articles in accordance with WHO FCTC recommendations would enhance the manuscript by addressing potential questions about the measures taken for adoption. (Incorporate this, if it is within the scope of your study)

4.      Consider providing direct quotes from the interviews in the main text rather than listing them in the reference list. This approach will offer readers immediate access to the interview content and enhance transparency in the sourcing of information. Also, if possible, mention the respondent's role while quoting a statement.

5.      Line number 193:” Following the EPHA’s journalist training sessions, health advocates noticed that journalists had established a coordinating committee network, consisting of members from different media institutions” and Line number 204” “that media was not well organized enough to push WHO FCTC-based legislation”. Both statements are from the same reference but contradict. Also in line 506, it is mentioned “The training of journalists helped create a deeper and well-coordinated network of local journalists that were able to constantly engage the government and  public on tobacco control issues”  

6.      Have a look into the below paper if useful.

·         A Media Analysis of the COVID-19 Tobacco Sales Ban in South Africa. https://doi.org/10.3390/ijerph20186733

·        Understanding the dynamics of notification and implementation of Article 5.3 across India’s states and union territories. https://tobaccocontrol.bmj.com/content/31/Suppl_1/s18

All the Best
